# Research on the Estimation of Chinese Fir Stand Volume Based on UAV-LiDAR Technology

**Shuhan Yu** [1,2] **, Xiangyu Chen** [1]**, Xiang Huang** [1]**, Yichen Chen** [1]**, Zhongyang Hu** [1]**, Jian Liu** [1] **and Kunyong Yu** [1,*]

[1]   College of Forestry, Fujian Agriculture and Forestry University, Fuzhou 350002, China;
     yushuhanjustin@163.com (S.Y.); xychen578@163.com (X.C.); huang991105@163.com (X.H.);
     15059080342@163.com (Y.C.); newtonhu@126.com (Z.H.); fjliujian@126.com (J.L.)
[2]   Fujian Chuanzheng Communications College, Fuzhou 350002, China
[*]   Correspondence: yuyky@126.com

**Abstract:** Chinese fir (*Cunninghamia lanceolata*) is the main fast-growing timber species in China, and studies of its stand volume are important for evaluating the effectiveness of forest management. However, it is difficult to accurately estimate stand volume from the perspective of a single tree due to the mutual concealment among Chinese fir trees. Therefore, in this study, we propose a method for identifying different forms of Chinese fir. The specific idea is to realize the accurate identification of a single Chinese fir tree, two Chinese fir trees, and three Chinese fir trees, and construct their respective stand volume estimation models to obtain an estimate of the forest stand volume. The key results are as follows: (1) the overall accuracy of recognition of different forms of Chinese fir is 79%, and the construction of different forms of Chinese fir units is beneficial for identifying forest trees; (2) a multiunit volume equation for different forms of Chinese fir is constructed; (3) based on predictions obtained with the constructed stand volume model, the difference between the estimated stand volume and the measured stand volume is small, and the average accuracy reaches 89.19%; and (4) compared to traditional volume estimation methods based on individual tree scale, the research method in this study shows a significant improvement (about 9.96%) in overall accuracy. In summary, this method can weaken the influence of erroneous individual tree segmentation on the accuracy of stand volume estimation, and can greatly reduce the working time of single tree segmentation to achieve the fast and accurate estimation of fir plantation stand volume.

**Keywords:** forest stand volume; Chinese fir; point cloud

## 1. Introduction

Forest stand volume refers to the total volume of all living standing trees in the forest, and it is one of the most important indicators in surveys of forest resources. This indicator can directly reflect the richness and scale of forest resources, and be used to measure and evaluate the carbon sequestration capacity of forests [1]. Moreover, it is an important parameter for identifying and evaluating the economic value of forest resources, and an important metric for evaluating forest management.

Traditional forest volume surveys mainly estimate forest volume through manual field measurements combined with statistical methods. The main methods include the method of mean tree and the volume table method [2]. This approach requires considerable manpower and material resources. It is relatively difficult to obtain a large range of forest parameters in a short period, and it is difficult to accomplish real-time dynamic monitoring of forest areas [3]. Since the 1970s, remote sensing technology has developed rapidly, and an increasing number of scholars have used remote sensing technology to monitor forest stands [4–6]. This technology provides technical support for large-scale and rapid forest stand estimation. In recent years, the rapid development of UAV-LiDAR technology has brought new opportunities for forest resource investigation, which is key

to the development of stand volume monitoring. Many scholars have employed UAV and LiDAR technology to study forest parameters and applied it in object extraction, tree species identification, individual tree segmentation, and stand volume estimation [7–12]. Constrained by the differences among trees and the randomness of the growth of trees in space, crowns can easily overlap and cover each other. As a result, it is difficult to segment point cloud data. Even in the case of individual tree segmentation, there are often erroneous cuts in clipping procedures, such as missed cuts and multiple cuts, which have a considerable impact on the extraction of single-tree information, and can greatly reduce the accuracy of forest stand volume estimation. Most scholars have addressed this problem by improving the quality of point cloud data and optimizing the single-tree segmentation method applied [13–16]. Determining how to identify single Chinese fir (*Cunninghamia lanceolata*) and other Chinese fir unit groups that cover each other in segmentation results, and how to calculate their respective stand volumes, are urgent problems to be solved. In this study, different forms of Chinese fir identification units are used to estimate stand volume and weaken the impact of incorrect individual tree segmentation on stand estimation accuracy.

Chinese fir is an important timber tree species in China, and is characterized by an early stage fast growth, strong adaptability, a large production volume, and low planting and management costs [17]. It is widely used and preferred for carbon sequestration and as a carbon sink [18]. Research on the stand volume of Chinese fir has practical application value for advancing the dual-carbon target within the forestry industry. Additionally, stand evaluation results can be used to assess the carbon sink capacity of Chinese fir forest resources at regional and larger scales; moreover, temporal and spatial trends can be assessed, and the results provide baseline data to support technical projects. Therefore, in this study, we focus on Chinese fir, use UAV airborne LiDAR point cloud data as the basic data source, construct an average tree height–canopy area stand volume model for different forms of Chinese fir, and combine the results of a high-precision comparison to obtain the optimal classification model for Chinese fir plantation stand volume estimation. This method can effectively prevent the problem of large errors in stand volume estimation due to erroneous individual tree segmentation results, and can provide a reference for future scholars to study forest stand volume, which is of great practical importance.

## 2. Study Area

The research area is in Yangkou State-owned Forest Farm (117°29′~118°14′ E, 26°38′~27°121′ N), Fujian Province, China (Figure 1). The average annual temperature is 18.5 °C, the average annual precipitation is 1880 mm, and the frost-free period is 305 days. With a large area of Krasnozem, it is suitable for the growth of fir and pine and other timber species. It is the center of China's fir production area [19,20].

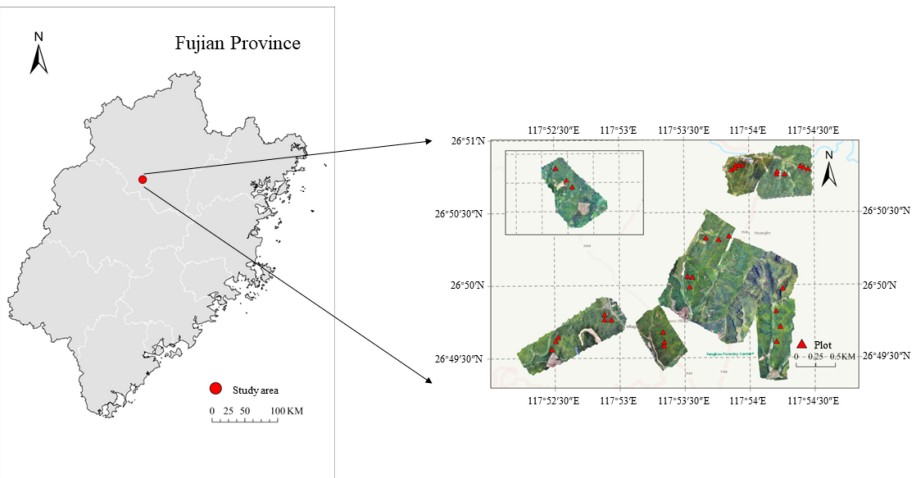

**Figure 1.** Location of study area.

## 3. Data Acquisition and Processing

### 3.1. Forest Stand Data Investigation and Volume Estimation

An investigation was conducted in the Chinese fir plantation area of Yangkou Forest Farm, Shunchang County, Nanping City, Fujian Province, on 22 August 2021. In total, 21 sampling plots, each with a dimension of 25.82 m × 25.82 m, were strategically arranged. The plots were systematically classified into two distinct groups: eleven plots (numbered 1–11) were allocated for accuracy assessment, while the remaining ten plots (numbered 12–21) were designated for model training. These selected plots encompassed Chinese fir plantations of varying age groups, ensuring a balanced representation across different stages of forest maturation. Real-time kinematic (RTK) was used to locate the plots and record all the center points and corner points of the sample plot, and to map individual trees in order to match UAV data to field measurements. A field investigation was conducted on 21 plots to obtain data on average age, average diameter at breast height (DBH), average tree height (TH), average crown width (CW), sum of basal area (BA), stand density, canopy closure, slope, aspect, and altitude (Table 1). Using the Fujian Province local standard DB35/T 1823-2019 tree volume formula of the main tree species in Fujian Province (see Equation (1)), the volume of each individual tree was calculated.

$$V = 0.0000706094 D^{1.801\,671} H^{0.997\,998} \tag{1}$$

where $V$ is cubic volume (m$^3$), $D$ is diameter at breast height (cm), and $H$ is tree height (m).

**Table 1.** Field investigation data of 21 plots.

| No. | Average Tree Age | Average DBH (cm) | Average TH (m) | Average CW (m) | BA (m$^2$) | Stand Density (Trees/hm) | Canopy Closure | Slope (°) | Aspect | Altitude (m) |
|---|---|---|---|---|---|---|---|---|---|---|
| 1 | 7 | 13.5 | 10 | 2.5 | 21,657.52 | 21,757.10 | 0.9 | 19 | W | 220 |
| 2 | 12 | 15.2 | 14 | 2.9 | 21,100.64 | 16,512.71 | 0.7 | 34 | W | 189 |
| 3 | 21 | 18.9 | 16 | 3.2 | 33,268.31 | 33,105.15 | 0.7 | 32 | E | 217 |
| 4 | 21 | 18.8 | 17 | 3.3 | 37,482.78 | 37,474.80 | 0.9 | 28 | EN | 227 |
| 5 | 25 | 19.5 | 17 | 3.2 | 31,563.16 | 22,398.57 | 0.6 | 30 | W | 255 |
| 6 | 31 | 20.9 | 17.2 | 3.6 | 31,668.11 | 31,562.42 | 0.7 | 36 | EN | 206 |
| 7 | 31 | 20.6 | 18 | 3.5 | 36,117.53 | 53,326.65 | 0.8 | 25 | EN | 247 |
| 8 | 29 | 19.1 | 16 | 3 | 27,350.23 | 42,691.64 | 0.9 | 30 | W | 209 |
| 9 | 58 | 27.2 | 19 | 6.5 | 29,157.34 | 29,053.45 | 0.6 | 32 | S | 246 |
| 10 | 56 | 28.9 | 22 | 6.8 | 28,802.31 | 28,862.79 | 0.6 | 32 | N | 202 |
| 11 | 56 | 29.1 | 19 | 7 | 26,690.92 | 26,603.32 | 0.6 | 28 | EN | 196 |
| 12 | 29 | 19.3 | 16 | 3.4 | 22,693.89 | 32,765.93 | 0.7 | 28 | W | 210 |
| 13 | 29 | 18.6 | 16.3 | 3.2 | 22,415.85 | 42,659.47 | 0.8 | 34 | WN | 211 |
| 14 | 12 | 16.9 | 12 | 3.1 | 18,903.87 | 18,842.68 | 0.7 | 26 | WN | 202 |
| 15 | 12 | 17.1 | 13.5 | 3 | 16,964.84 | 16,994.71 | 0.6 | 23 | EN | 196 |
| 16 | 21 | 19.2 | 16 | 2.7 | 37,329.38 | 37,349.26 | 0.8 | 30 | E | 217 |
| 17 | 25 | 22.3 | 18 | 3.4 | 29,258.12 | 42,318.33 | 0.9 | 16 | WS | 223 |
| 18 | 37 | 31.3 | 22 | 5.5 | 36,203.6 | 36,163.99 | 0.6 | 17 | WS | 220 |
| 19 | 7 | 13 | 8.5 | 2.6 | 21,394.10 | 19,731.98 | 0.7 | 26 | WN | 217 |
| 20 | 7 | 12.6 | 9 | 2.5 | 25,715.00 | 24,065.13 | 0.9 | 26 | W | 230 |
| 21 | 37 | 33.8 | 22 | 6.6 | 35,934.74 | 35,890.81 | 0.8 | 16 | W | 308 |

Then, the stand volume value of each plot was obtained, and was used as the input for the accuracy verification of the stand volume estimation model in the later stage.

### 3.2. Remote Sensing Data Collection and Preprocessing

The DJI Phantom 4 UAV with an integrated multispectral imaging system had a heading overlap rate of 80%, a side overlap rate of 80%, a flight height of 284 m, and a spatial resolution of 15 cm. LiDAR point cloud data were also collected at the same time using a Pegasus LiDAR UAV D500 equipped with a HESAI XT32 sensor to obtain

high-density point cloud data. We chose to implement take-off at noon, when the weather was clear, to minimize the influence of the shadow of the fir tree crown. Considering the large terrain fluctuation of the plots (the slope of the plots ranges from 16° to 36°), the flight mode adopted the ground imitation flight mode, the ground imitation height was set to 150 m, the overlap rate was set to 80%, the three echo mode was adopted, and the laser level was CLASS1. Ten plots of different age groups were selected as the sample set, and CloudCompare_v2.10 software was used to perform preprocessing, such as point cloud stitching and SOR (statistical outlier removal) filter denoising, on the airborne LiDAR point cloud.

## 4. Remote Sensing Analysis

In this study, the PCS algorithm was used to segment the UAV LiDAR point cloud data of 10 plots of different age groups (No. 12–21), and then two-dimensional processing was performed on the segmentation results. In the results, there are cases in which a single Chinese fir was not covered, two Chinese firs were covered by each other, and three Chinese firs were covered by each other. Therefore, we defined them as recognition units of different forms based on the ground survey data (Figure 2). Based on the sample sets of the recognition units of different forms, Chinese fir stand volume models were constructed based on the average tree height–canopy area of different forms.

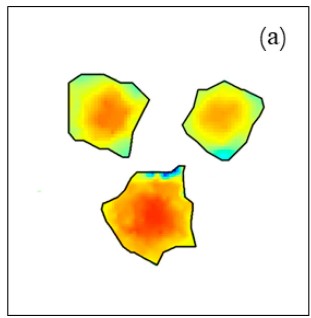 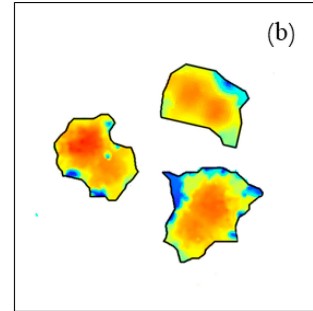 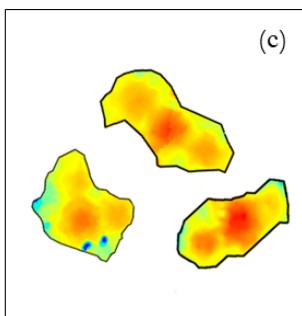

**Figure 2.** Chinese fir recognition units of different forms. (**a**) A single Chinese fir tree of recognition units; (**b**) Two Chinese fir trees of recognition units; (**c**) Three Chinese fir trees of recognition units.

The optimal recognition model was screened out according to the spatial characteristics of the recognition sample set, and the tree heights of different recognition units were extracted using the K-means algorithm. Another 11 plots (No. 1–11) of different age groups were selected for accuracy verification to ensure the applicability and reliability of the model (Figure 3). Finally, the stand volume obtained using this research method was compared with the stand volume calculated using the traditional single-tree segmentation method.

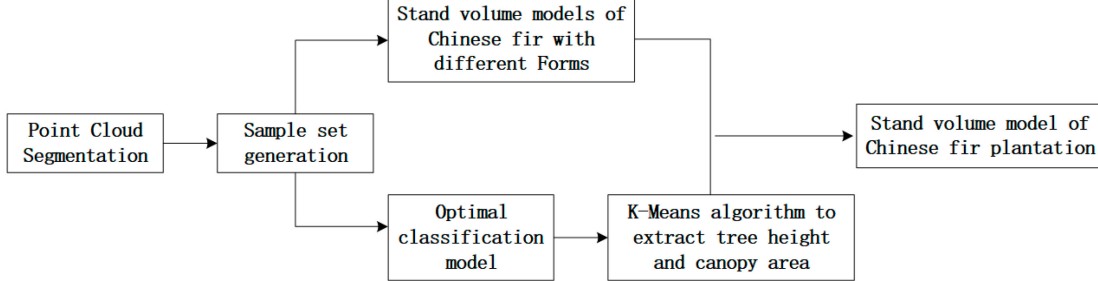

**Figure 3.** Flow chart of construction of a volume model of a Chinese fir plantation.

### 4.1. Individual Tree Segmentation

After obtaining the UAV LiDAR point cloud data of 10 plots, it is necessary to quickly and accurately divide the ground points and nonground points. Most of the traditional filtering algorithms consider the difference between slope and elevation changes to distinguish ground object points from ground points, while the cloth filtering algorithm (cloth

simulation filter) performs filtering using a completely new approach [21]. The main idea is to first flip the point cloud and then assume that a piece of cloth falls freely from above due to gravity, and the area covered by the cloth is the current ground point cloud. After separating the ground point cloud and nonground point cloud of the 10 plots in the study area through the cloth filter algorithm (CSF), ArcMap 10.8 software was used to convert the LAS dataset into a raster with 15 cm spatial resolution, extract the digital surface model (DSM) and digital elevation model (DEM) of all plots, and then subtract them to obtain the vegetation canopy height model (CHM), which was used to assist in the construction of Chinese fir recognition units.

In this study, the PCS algorithm was used to perform single-tree segmentation on the point cloud data of 10 plots (Figure 4). The PCS algorithm (point cloud segmentation) is mainly based on the relative spacing between trees, and a single tree is sequentially segmented from the point cloud [22]. The algorithm first assumes that the highest point is the height of the tree, grows the area from the tree vertex, excludes the points of other trees according to the relative distance, and finally determines whether to split a tree. In this way, one tree is split in each iteration until all trees are split.

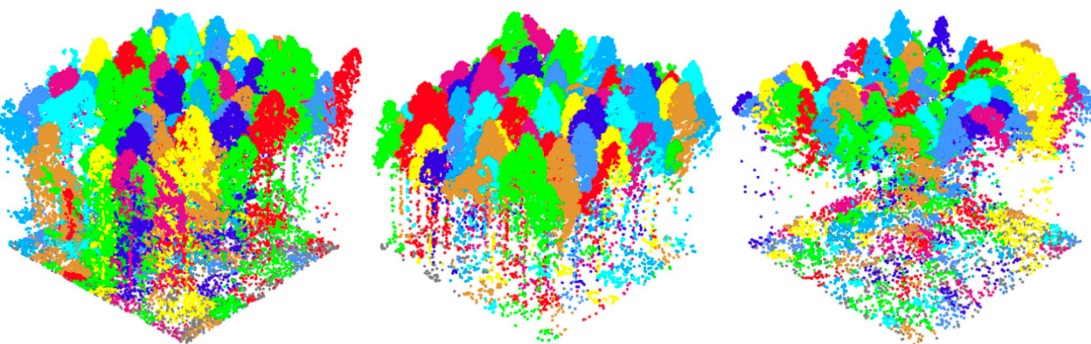

**Figure 4.** Point cloud segmentation results of some plots under the PCS algorithm.

## 4.2. Recognition Unit Construction and Feature Extraction

The alpha shape algorithm proposed by Edelsbrunner H [23] is a simple and effective method for quickly extracting boundary points; it overcomes the shortcomings of the influence of the shape of point cloud boundary points and can quickly and accurately extract boundary points. The alpha shape algorithm was used to extract the point cloud boundary points from the segmentation results of 10 plots (Figure 5), and after forming the SHP files, the mask was extracted from the multispectral images of the 10 plots to obtain the final two-dimensional grid segmentation results. The alpha shape algorithm was used to extract point cloud boundary points from the segmentation results of 10 plots, and after forming vector data, mask extraction was performed on the multispectral images of these 10 plots to obtain the final two-dimensional raster segmentation results. Combined with plot survey data, multispectral image data and the canopy height model, 99 single Chinese fir tree, 97 two Chinese fir tree, and 71 three Chinese fir tree recognition unit samples were finally selected. A sample set of recognition units was constructed using visual interpretation.

The purpose of individual tree segmentation is to obtain objects for classification, and the recognition unit is also a sample for target recognition. Geometric features are unique feature parameters of object-oriented information extraction that can quantitatively describe the shape and area of actual ground objects. Different forms of Chinese fir canopy can be represented by geometric features such as area, length, compactness, convexity, solidity, roundness, and form factor (Table 2). These geometric feature metrics were used as predictors for classification models identifying different forms of recognition units. Considering the geometric feature value in the recognition unit as the basic data source, the accuracy of the six classification models is compared, and finally, the optimal classification model can be obtained.

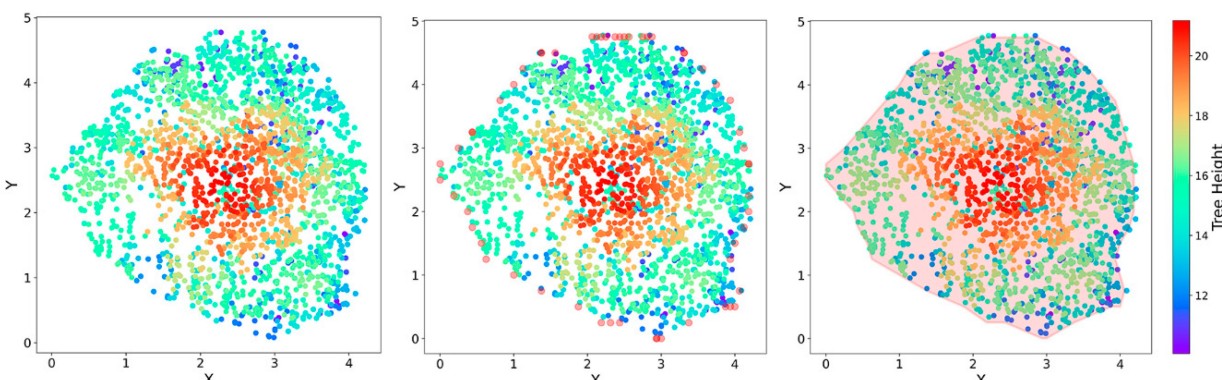

**Figure 5.** Alpha-shapes algorithm used to extract point cloud boundary points.

**Table 2.** Geometric feature description.

| Geometric Feature | Description |
|---|---|
| Area | Total area of the polygon, minus the area of the holes |
| Length | The combined length of all boundaries of the polygon, including the boundaries of the holes |
| Compactness | A shape measure that indicates the compactness of the polygon. A circle is the most compact shape with a value of 1/pi. The compactness value of a square is 1/2(sqrt(pi)). Compactness = Sqrt (4×area/pi)/outer contour length |
| Convexity | Polygons are either convex or concave. This attribute measures the convexity of the polygon. The convexity value for a convex polygon with no holes is 1.0, while the value for a concave polygon is less than 1.0 Convexity = length of convex hull/length |
| Solidity | A shape measure that compares the area of the polygon to the area of a convex hull surrounding the polygon. The solidity value for a convex polygon with no holes is 1.0, and the value for a concave polygon is less than 1.0 Solidity = area/area of convex hull |
| Roundness | A shape measure that compares the area of the polygon to the square of the maximum diameter of the polygon. The "maximum diameter" is the length of the major axis of an oriented bounding box enclosing the polygon. The roundness value for a circle is 1, and the value for a square is 4/pi Roundness = 4× (area)/(pi× Major_Length$^2$) |
| Form_Factor | A shape measure that compares the area of the polygon to the square of the total perimeter. The form factor value of a circle is 1, and the value of a square is pi/4 Form_Factor = 4× pi×(area)/(total perimeter)$^2$ |

### 4.3. Tree Volume Model Screening

By calculating the average tree height and canopy area of a single Chinese fir tree, two Chinese fir trees and three Chinese fir tree sample sets, the calculated individual tree volume values were combined. Six commonly used tree volume models (Table 3) were selected, and SPSS Statistics V26 statistical analysis software was used to test the correlation between parameters. Finally, the optimal tree volume models of Chinese firs in different forms were determined. We accumulated the calculated tree volumes of different

recognition units, and then divided the final result by the area of the corresponding sample plot to obtain the final plot stand volume (see Equation (2)).

$$M = \frac{\sum_{i=1}^{n} V_i}{S} \tag{2}$$

where $M$ is stand volume ($m^3 \cdot hm^{-2}$), $V_i$ is the tree volume of the $i$-th recognition unit ($m^3$), $S$ is plot area ($m^2$), and $n$ is total number of recognition units.

**Table 3.** Tree volume model.

| Model Number | Model Calculation Formula |
|:---:|:---:|
| 1 | $V = a_0 + a_1 D + a_2 D^2 + a_3 DH + a_4 D^2 H + a_5 H$ |
| 2 | $V = a_0 D^{a_1} H^{a_2}$ |
| 3 | $V = a_0 + a_1 D^2 + a_2 D^2 H + a_3 H$ |
| 4 | $V = a_0 + a_1 D^2 + a_2 D^2 H + a_3 H + a_4 DH^2$ |
| 5 | $V = a_0 (D + 1)^{a_1} H^{a_2}$ |
| 6 | $V = a_0 (H/D)^{a_1} D^2 H$ |

The selected optimal classification model was used to classify and identify the other 11 plots, and the accuracy of the acquired fir recognition units of different shapes was verified via visual interpretation. The K-means algorithm [24] was used to find the tree vertices from the recognition results, combined with the vegetation canopy height model (CHM), to extract the average tree height, which was used to calculate the average tree height–canopy area stand volume model of Chinese firs in different forms. The actual stand volume of 11 plots was calculated through ground survey data, and the accuracy was verified with the estimated stand volume.

## 5. Result

### 5.1. Accuracy Evaluation of Different Classification Models

We selected 70% of the 267 recognition unit samples as the training set, and 30% of the samples as the validation set. A total of six classification models, including decision tree, k-nearest neighbor, support vector machine, random forest, perceptron and BP neural network, were selected for accuracy evaluation; the random forest classification model had the highest accuracy, with an overall accuracy of 79% (Table 4). This result demonstrated that the random forest model is ideal for the accurate identification of different forms of Chinese fir, and it can be used to identify and classify plots.

**Table 4.** Accuracy evaluation of different classification models.

| Machine Learning Model | Overall Accuracy |
|:---:|:---:|
| Decision Tree | 72% |
| k-Nearest Neighbor | 73.95% |
| Support Vector Machine | 75% |
| Random Forest | 79% |
| Perceptron | 52.81% |
| BP Neural Network | 69% |

### 5.2. Construction of Chinese Fir Stand Volume Models in Different Forms

5.2.1. Construction of the Tree Height–Canopy Area Tree Volume Model of a Single Chinese Fir Tree

The parameter settings of each model are shown in Table 5, Table 6 shows that the $R^2$ of model 4 was the largest, with a value of 0.9543. Its RMSE was also the smallest (0.0280 $m^3$), indicating that it was better than that of the other five models. Model 4 has the

best fitting results among the six tree volume alternative models, and it was determined to be the optimal model. The equation is as follows.

$$V = 0.00221993 - 0.00004764S^2 - 0.00005463S^2H + 0.00651276H + 0.00007023SH^2 \quad (3)$$

where $V$ is the tree volume of the plots ($m^3$), $S$ is the canopy area of a single Chinese fir tree ($m^2$), and $H$ is the height of a single Chinese fir (m).

**Table 5.** Parameters of the tree height–canopy area tree volume model of a single Chinese fir tree.

| Model Number | Model Parameters | | | | | |
|---|---|---|---|---|---|---|
| | $a_0$ | $a_1$ | $a_2$ | $a_3$ | $a_4$ | $a_5$ |
| 1 | 0.00114022 | 0.00374371 | −0.00252036 | 0.002006 | 0.00003603 | 0.002276 |
| 2 | 0.00064871 | 0.00153772 | 2.05522234 | — | — | — |
| 3 | −0.11748344 | −0.00114966 | 0.00005589 | 0.021143 | — | — |
| 4 | 0.00221993 | −0.00004764 | −0.00005463 | 0.006513 | 0.00007023 | — |
| 5 | 0.00065078 | 0.00600983 | 2.05034974 | — | — | — |
| 6 | 0.00003101 | 2.19820343 | — | — | — | — |

**Table 6.** Evaluation of the tree height–canopy area tree volume model of a single Chinese fir tree.

| Model Number | Model Evaluation Index | | | Result Order |
|---|---|---|---|---|
| | $R^2$ | RSS | RMSE | |
| 1 | 0.9469 | 0.0896 | 0.0302 | 2 |
| 2 | 0.9427 | 0.0967 | 0.0314 | 3 |
| 3 | 0.9297 | 0.1186 | 0.0348 | 4 |
| 4 | 0.9543 | 0.0771 | 0.0280 | 1 |
| 5 | 0.9427 | 0.0967 | 0.0314 | 3 |
| 6 | 0.8124 | 0.3163 | 0.5681 | 5 |

### 5.2.2. Construction of the Average Tree Height–Canopy Area Tree Volume Model of Two Chinese Fir Trees

The parameter settings of each model are shown in Table 7, Table 8 shows that the $R^2$ of model 2 was the largest, at 0.9490. Its RMSE was also the smallest (0.0591 $m^3$), indicating that it was better than that of the other five models. Model 2 had the best fitting results among the six tree volume alternative models, and it was determined to be the optimal model. The equation is as follows.

$$V = 0.00124227S^{-0.19385244}H^{2.28001331} \quad (4)$$

where $V$ is the tree volume of the plots ($m^3$), $S$ is the canopy area of two Chinese fir trees ($m^2$), and $H$ is the average tree height of two Chinese fir trees (m).

**Table 7.** Parameters of the average tree height–canopy area tree volume model of two Chinese fir trees.

| Model Number | Model Parameters | | | | | |
|---|---|---|---|---|---|---|
| | $a_0$ | $a_1$ | $a_2$ | $a_3$ | $a_4$ | $a_5$ |
| 1 | −0.15937766 | −0.01361136 | −0.00036700 | 0.00072852 | 0.00001908 | 0.03944424 |
| 2 | 0.00124227 | −0.19385244 | 2.28001331 | — | — | — |
| 3 | −0.24074940 | −0.00078606 | 0.00004082 | 0.04387297 | — | — |
| 4 | −0.12736666 | −0.00019427 | −0.00002202 | 0.02760795 | 0.00006037 | — |
| 5 | 0.00130378 | −0.20312478 | 2.27671086 | — | — | — |
| 6 | 0.00007405 | 2.34306418 | — | — | — | — |

**Table 8.** Evaluation of the average tree height–canopy area tree volume model of two Chinese fir trees.

| Model Number | Model Evaluation Index | | | Result Order |
| --- | --- | --- | --- | --- |
| | $R^2$ | RSS | RMSE | |
| 1 | 0.9352 | 0.4263 | 0.0666 | 4 |
| 2 | 0.9490 | 0.3357 | 0.0591 | 1 |
| 3 | 0.9344 | 0.4318 | 0.0671 | 5 |
| 4 | 0.9426 | 0.3778 | 0.0627 | 3 |
| 5 | 0.9488 | 0.3368 | 0.0592 | 2 |
| 6 | 0.8180 | 1.1978 | 0.1117 | 6 |

5.2.3. Construction of the Average Tree Height–Canopy Area Tree Volume Model of Three Chinese Fir Trees

The parameter settings of each model are shown in Table 9, Table 10 shows that the $R^2$ of model 1 was the largest, at 0.9068. Its RMSE was also the smallest (0.1243 m$^3$), indicating that it was better than that of the other five models. Model 1 had the best fitting results among the six tree standing volume alternative models, and it was determined to be the optimal model. The equation is as follows.

$$V = -0.22361391 - 0.00909546S - 0.00042003S^2 + 0.00100871SH + 0.00001353S^2H + 0.05011857H \tag{5}$$

where $V$ is the tree volume of the plots (m$^3$); $S$ is the canopy area of three Chinese fir trees (m$^2$); and $H$ is the average tree height of three Chinese fir trees (m).

**Table 9.** Parameters of the average tree height–canopy area tree volume model of three Chinese fir trees.

| Model Number | Model Parameters | | | | | |
| --- | --- | --- | --- | --- | --- | --- |
| | $a_0$ | $a_1$ | $a_2$ | $a_3$ | $a_4$ | $a_5$ |
| 1 | −0.22361391 | −0.00909546 | −0.00042003 | 0.00100871 | 0.00001353 | 0.05011857 |
| 2 | 0.00220577 | 0.03562318 | 1.99310134 | — | — | — |
| 3 | −0.35694824 | −0.00051649 | 0.00002714 | 0.06452083 | — | — |
| 4 | −0.29275905 | −0.00038723 | 0.00001322 | 0.05569801 | 0.00002123 | — |
| 5 | 0.00219217 | 0.03862131 | 1.99135957 | — | — | — |
| 6 | 0.00012336 | 2.22696342 | — | — | — | — |

**Table 10.** Evaluation of the average tree height–canopy area tree volume model of three Chinese fir trees.

| Model Number | Model Evaluation Index | | | Result Order |
| --- | --- | --- | --- | --- |
| | $R^2$ | RSS | RMSE | |
| 1 | 0.9068 | 1.0824 | 0.1243 | 1 |
| 2 | 0.8999 | 1.1625 | 0.1289 | 4 |
| 3 | 0.9057 | 1.0949 | 0.1251 | 3 |
| 4 | 0.9067 | 1.0832 | 0.1244 | 2 |
| 5 | 0.8999 | 1.1625 | 0.1289 | 4 |
| 6 | 0.7835 | 2.5149 | 0.1895 | 5 |

*5.3. Applicability Evaluation of the Chinese Fir Plantation Stand Volume Model*

The random forest model was selected to classify and identify 11 plots, and 296 single Chinese fir trees, 272 groups of two Chinese fir trees, and 112 groups of three Chinese fir trees were obtained. Combined with the ground survey data for accuracy verification, the results show that the identification accuracy of a single Chinese fir is the highest, at 97.9%; the identification accuracy of three Chinese firs is the lowest, at 82.11%; the overall identification accuracy reaches 91.66% (Table 11). The K-means algorithm and PCS algorithm were used to extract the average tree height and canopy area, and combined with the optimal stand volume equation, the stand volume estimation results of the Chinese fir plantation were finally obtained. Comparing the estimated stand volume of all plots

with the actual stand volume, the highest precision was 98.61%, the lowest precision was 78.86%, and the average precision was 89.19% (Table 12).

**Table 11.** The identification results of different forms of Chinese fir in 11 plots.

| Different Forms of Chinese Fir Recognition Units | Predicted Number | Actual Number | Relative Error (%) | Precision (%) |
|---|---|---|---|---|
| Single Tree | 296 | 290 | 2.1 | 97.90 |
| Two Trees | 272 | 259 | 5.02 | 94.98 |
| Three Trees | 112 | 95 | 17.89 | 82.11 |

**Table 12.** Predicted and actual values of Chinese fir quantity and stand volume in the plots.

| Plot Number | Actual Quantity of Chinese Fir Trees | Predicted Quantity of Chinese Fir Trees | Actual Stand Volume ($m^3 \cdot hm^{-2}$) | Predicted Stand Volume ($m^3 \cdot hm^{-2}$) | Stand Volume Precision (%) |
|---|---|---|---|---|---|
| 1 | 152 | 176 | 169.57 | 133.73 | 78.86 |
| 2 | 91 | 105 | 214.69 | 199.4 | 92.88 |
| 3 | 118 | 113 | 400 | 394.45 | 98.61 |
| 4 | 135 | 126 | 482.91 | 427.14 | 88.45 |
| 5 | 62 | 89 | 393.4 | 427.44 | 91.35 |
| 6 | 92 | 111 | 380.36 | 416.19 | 90.58 |
| 7 | 160 | 148 | 490.85 | 449.78 | 91.63 |
| 8 | 149 | 132 | 343.63 | 284.86 | 82.90 |
| 9 | 50 | 64 | 371.51 | 314.24 | 84.58 |
| 10 | 44 | 61 | 436.28 | 385.31 | 88.32 |
| 11 | 40 | 51 | 350.07 | 374.81 | 92.93 |

With the quantity of Chinese fir trees measured in the plots as the independent variable x, and the quantity of Chinese fir trees predicted by the model as the dependent variable y, a linear regression equation was established: y = 0.7972x + 27.693, where $R^2$ = 0.8984 (Figure 6). With the stand volume obtained from the actual measurement of the plots as the independent variable x, and the stand volume estimated by the model as the dependent variable y, a linear regression equation was established: y = 0.9608x − 6.1666, where $R^2$ = 0.8671 (Figure 7).

Residual analysis was carried out between the measured value of the storage volume and the estimated value of the model constructed in the study (Figure 8), and the results showed that the residuals were evenly distributed on both sides of Y = 0 and were between ±1.5. Therefore, the linear model constructed in the study is statistically significant. Then, the paired sample T test was carried out on the measured value and the estimated value (Table 13); the result showed that the T value was 1.82, and the significance probability $p > 0.05$. Therefore, the null hypothesis is rejected, indicating that there is no significant difference between the measured value and the estimated value of the accumulation, that is, the accumulation prediction model constructed in this study has the expected effect.

**Table 13.** Paired sample test of the predicted and actual values of the stand volume of the plots.

| | Component Difference | | | | |
|---|---|---|---|---|---|
| Average | Standard Deviation | Standard Error | T | df | *p* |
| 20.54 | 37.54 | 11.32 | 1.82 | 10 | 0.1 |

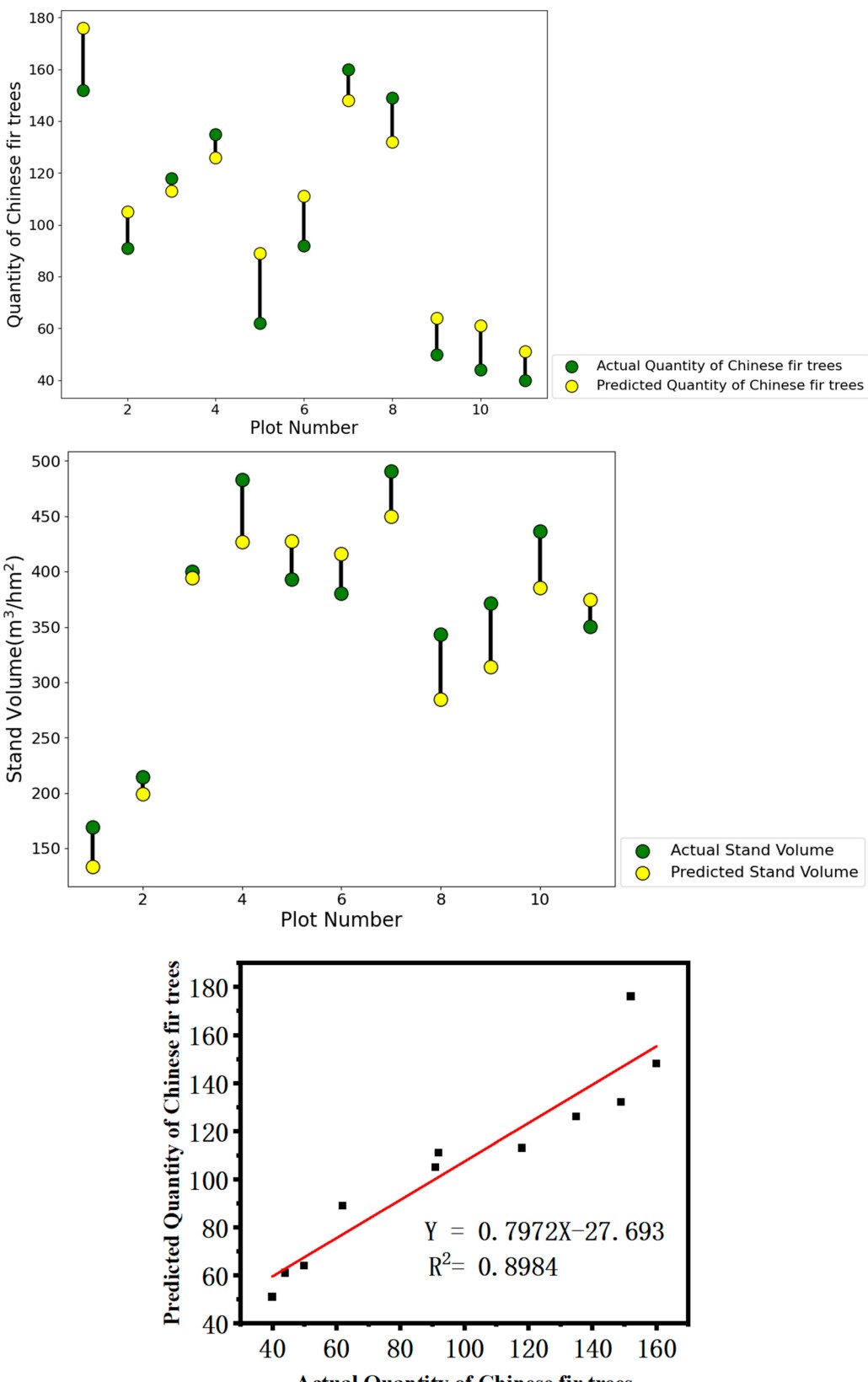

**Figure 6.** Linear regression between the actual and model-predicted quantity of Chinese fir trees in the plots.

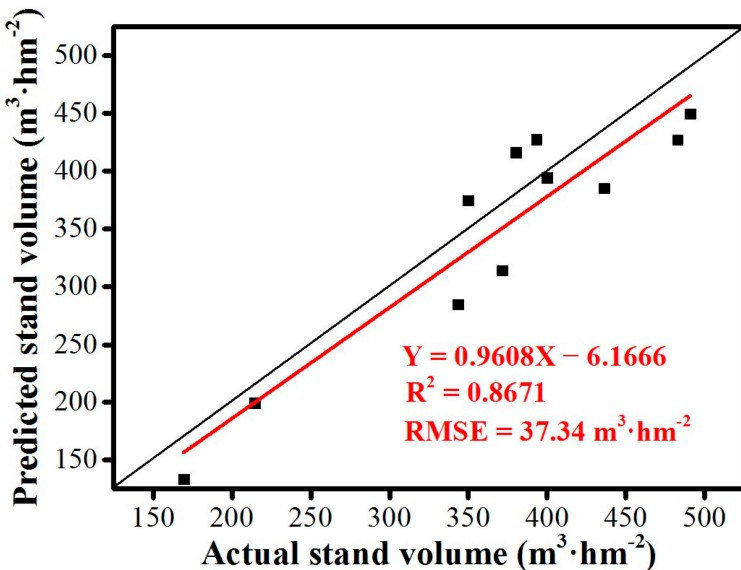

**Figure 7.** Linear regression between the actual and the model-predicted values of the Chinese fir stand volume of the plots.

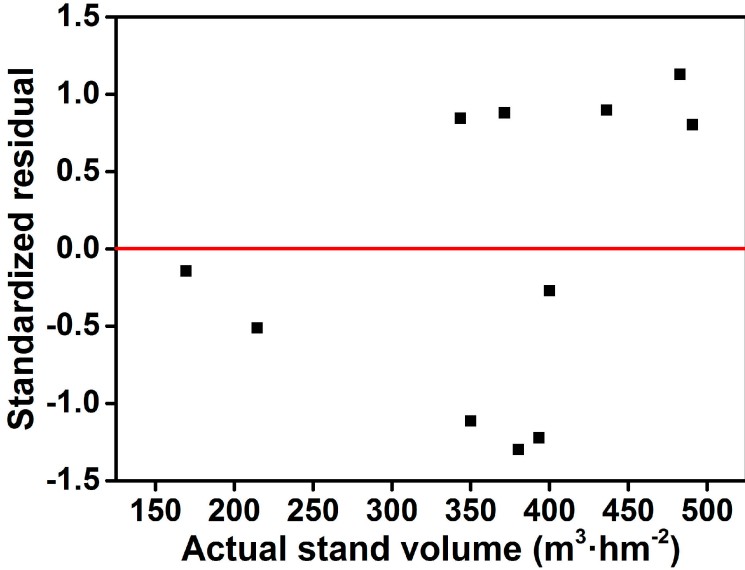

**Figure 8.** Residual distribution of Chinese fir stand volume.

*5.4. Prediction of Stand Volume Based on Individual Tree Scale*

The PCS algorithm was used to segment 11 sample plots (No. 1–11); 213 individual tree samples were selected as the basic data to estimate the model parameters, and the correlation test between the parameters was carried out. The parameter settings of each model are shown in Table 14, Table 15 shows that the $R^2$ of model 1 was the largest, at 0.856. Its RMSE was also the smallest (0.118 m$^3$), indicating that it was better than that of the other five models. Model 1 had the best fitting results among the six tree standing volume alternative models, and it was determined to be the optimal model. The equation is as follows.

$$V = 0.17437123 - 0.00860561S - 0.00074231S^2 + 0.00367536SH - 0.00000621S^2H - 0.01908210H \quad (6)$$

where $V$ is the tree volume of the plots (m$^3$); $S$ is the canopy area of Chinese fir trees (m$^2$); and $H$ is the tree height of Chinese fir tree (m).

**Table 14.** Parameters of the tree height–canopy area tree volume model at the single-tree scale.

| Model Number | Model Parameters | | | | | |
|---|---|---|---|---|---|---|
| | $a_0$ | $a_1$ | $a_2$ | $a_3$ | $a_4$ | $a_5$ |
| 1 | 0.17437123 | −0.00860561 | 0.00074231 | 0.00367536 | −0.00000621 | −0.01908210 |
| 2 | 0.00132772 | 0.65890593 | 1.43467226 | — | — | — |
| 3 | −0.21253527 | 0.00013146 | 0.00001556 | 0.03354285 | — | — |
| 4 | 0.12139797 | 0.00057969 | −0.00006051 | −0.00706735 | 0.00012632 | — |
| 5 | 0.00115235 | 0.69485526 | 1.43276099 | — | — | — |
| 6 | 0.00009124 | 1.55492197 | — | — | — | — |

**Table 15.** Evaluation of the tree height–canopy area tree volume model at the single-tree scale.

| Model Number | Model Evaluation Index | | | Result Order |
|---|---|---|---|---|
| | $R^2$ | RSS | RMSE | |
| 1 | 0.856 | 2.935 | 0.118 | 1 |
| 2 | 0.835 | 3.372 | 0.126 | 3 |
| 3 | 0.781 | 4.476 | 0.145 | 5 |
| 4 | 0.841 | 3.247 | 0.124 | 2 |
| 5 | 0.834 | 3.396 | 0.127 | 4 |
| 6 | 0.755 | 4.995 | 0.153 | 6 |

We added up the tree volume of all individual trees, and then divided the final result by the corresponding sample plot area to obtain the final sample plot stand volume. At the single-tree scale, the highest precision was 95.47%, the lowest precision was 56.02%, and the average precision was 79.23% when comparing the estimated forest stand volume with the actual forest stand volume in 11 plots (Table 16).

**Table 16.** Predicted and actual values of Chinese fir stand volume in 11 plots at the single-tree scale.

| Plot Number | Actual Stand Volume ($m^3 \cdot hm^{-2}$) | Predicted Stand Volume ($m^3 \cdot hm^{-2}$) | Stand Volume Precision (%) |
|---|---|---|---|
| 1 | 169.57 | 195.49 | 84.72 |
| 2 | 214.69 | 224.41 | 95.47 |
| 3 | 400 | 325.54 | 81.38 |
| 4 | 482.91 | 347.65 | 71.99 |
| 5 | 393.4 | 332.40 | 84.49 |
| 6 | 380.36 | 346.66 | 91.14 |
| 7 | 490.85 | 274.98 | 56.02 |
| 8 | 343.63 | 193.91 | 56.43 |
| 9 | 371.51 | 314.92 | 84.77 |
| 10 | 436.28 | 390.22 | 89.44 |
| 11 | 350.07 | 435.12 | 75.71 |

## 6. Discussion

Within this study, we have explored, categorized, and discussed individual, two, and three Chinese fir trees. This has enabled us to construct optimal stand volume models for different Chinese fir formations, facilitating rapid and accurate estimations of the stand volume of Chinese fir plantations. The main advantages of this research method are as follows. (1) In the case of forest stands that cannot be accurately segmented due to high canopy density, this method does not require the accurate segmentation of each individual tree, and can also obtain better results of forest stand volume. Therefore, the method can save time for optimizing segmentation results, and can greatly improve the efficiency of forestry work. (2) Compared with the traditional DBH-tree height volume model, this method does not require much manpower or material resources to carry out the work of measuring each tree to obtain DBH, tree height and other information. The information required for the stand volume model based on the average tree height-canopy area can be quickly obtained using UAV-LiDAR technology alone, which can reduce the amount of

fieldwork needed, saving time for researchers and minimizing the risks associated with outdoor work. (3) The average accuracy of volume estimation based on the individual tree scale stands at 79.23%. However, using the methodology proposed in our research, the average accuracy can be elevated to 89.19%. This demonstrates that our proposed method effectively mitigates the accuracy loss in volume estimation results, a common issue triggered by erroneous individual tree segmentation in forest stands.The main shortcoming of this study is: Compared with the traditional DBH–tree height stand volume model, the stand volume model constructed in this study had a slightly lower overall accuracy. This is because the former model has been researched and discussed by many scholars, and the selection and fitting of the model is more mature. When the model ($M = aH^bN^c + a_1H^{b1}N^{c1}$) constructed by Kang L [25] was applied to research on the stand volume of middle-aged Chinese fir forests in the state-owned forest farms of Hunan Province in China, the accuracy was 97.1948%. Lu Y [26] used the method of a standard volume table to estimate stand volume based on the DBH and TH of Chinese firs, with an accuracy of 97.95%. In the future, we will continue to optimize the accuracy of the recognition units to further improve the accuracy of the accumulation estimation model.

Table 11 clearly indicates a descending trend in the recognition accuracy of the unit, as the number of Chinese firs involved in the formation of the recognition unit increases. This observation could potentially be attributed to the varying degrees of shading and the diverse forms of shading among the crowns of Chinese fir trees, as previously documented in the literature [27–31]. The increased number of overlapping Chinese firs leads to an expanded range of possible coverage combinations, resulting in significant variations in canopy area, thereby influencing the overall tree crown identification accuracy. Future investigations will aim to explore the form and degree of coverage among Chinese firs from a multi-perspective approach, in order to understand the effect of different coverage conditions on the stand volume model's accuracy.

Airborne LiDAR has its limitations, especially in capturing vertical information, and the mutual occlusion between Chinese fir crowns might result in missing lower canopy point cloud data, thus hindering us from acquiring a complete and real crown of the fir trees. We plan to overcome these challenges by combining UAV and Terrestrial Laser Scanning (TLS) to obtain comprehensive single tree point cloud data in future work, with the goal of improving the accuracy of the accumulation model.

Based on the machine learning method, in this study, we used multiple geometric features to construct recognition units of different forms of Chinese fir. Compared with Zhang Canghao [32], the selected indicators are more comprehensive and can better reflect the canopy shapes of different recognition units. In this study, we found that the highest classification accuracy was achieved via the random forest methodology. Some authors have noted that due to its strong anti-noise ability and high stability, the random forest methodology is the most widely used in the classification research of forestry and agriculture [33,34]. It can usually achieve greater classification accuracy and can classify and count high-dimensional data while ensuring the randomness of samples. Compared with other classifiers, the random forest classifier has lower requirements for data sources and can obtain better results even if there is a lack of information. Although it has the characteristics of strong universality, the introduction of geometric features reduced the classification accuracy [35,36], so the final classification accuracy in this study was 79%, which was slightly lower. For this reason, the introduction of spectral features, index features, and texture features must be considered in follow-up research to further enhance the classification of recognition units, thereby improving the accuracy of the stand volume model.

After predicting the stand volume of 11 plots, the difference between the predicted value and the actual value was small, and the average precision was high. However, there were still a few sample plots with large differences, and the main reasons are as follows: (1) The ground undulations in part of the forest are large, and the microtopography of the ground obviously leads to the estimation error of the tree height. At the same time, tending efforts on young forests are relatively small, resulting in large canopy density

and sparse ground point clouds, which also lead to inaccurate tree height extraction. (2) During the natural growth of forest trees, as the competition pressure between different tree individuals continues to increase, the phenomenon of natural pruning also continues to increase, weakening the typicality of the morphological characteristics of the canopy identification unit, which, in turn, affects the estimation of forest stand volume.

In this study, we implemented rapid estimation of the stand volume of Chinese fir plantations by establishing stand volume models of different forms of Chinese fir. The experimental results also demonstrated that the model can indeed be used for forest resource monitoring of Chinese fir plantations, and has strong forestry service capabilities. However, this study is limited to the estimation of the stand volume of Chinese fir plantations, and its model has not been used and verified for other tree species. Whether it has good applicability in the stock volume estimation of other tree species remains to be determined.

## 7. Conclusions

In the monitoring of forest resources, the determination of forest stand volume requires high manpower and material costs. In forest stands with high canopy density, there are phenomena such as mutual occlusion and adhesion between individual tree canopies, which lead to difficulties in extracting information about the number of trees and individual tree canopies in high canopy density stands. Although there are many individual tree segmentation technologies that can be used to perform remote sensing estimation of forest volume; generally, optimizing and debugging the target forest segmentation results requires considerable time. However, this study demonstrates that there is a significant correlation between the average tree height, canopy area, and volume of different forms of Chinese fir, and the obtained stock model had a high accuracy, with $R^2$ values greater than 0.9.

In this study, the abovementioned stand volume model was used to predict 11 plots. The results showed that the predicted stand volume of the model was relatively close to the actual stand volume value, and the average accuracy was greater than 90%. Compared to traditional volume estimation methods based on individual tree scale, the research method in this study shows a significant improvement (about 9.96%) in overall accuracy, which helps mitigate errors in stand volume estimation caused by incorrect individual tree segmentation. Therefore, the average tree height–canopy area stand volume model of different forms of Chinese fir constructed in this study exhibited strong feasibility. Forestry workers can use this model to quickly estimate the stand volume of Chinese fir plantations, saving the processing time required for individual tree division. It has strong practical importance and can be used as the basis of future research on forest stand volume estimation in Chinese fir plantations.

**Author Contributions:** Conceptualization, S.Y.; Methodology, S.Y.; Software, X.C.; Investigation, X.C., X.H., Y.C. and Z.H.; Data curation, X.H.; Writing—original draft, S.Y.; Writing—review & editing, S.Y.; Supervision, K.Y.; Project administration, J.L. and K.Y.; Funding acquisition, K.Y. All authors have read and agreed to the published version of the manuscript.

**Funding:** This work was supported by Research on Key technologies of intelligent monitoring and carbon sink metering of forest resources in Fujian Province (grant number 2022FKJ03), National Natural Science Foundation Project (grant number 32271876) and Science and technology plan project of Fujian Provincial Department of Water Resources (grant number MSK202106).

**Institutional Review Board Statement:** Not applicable.

**Informed Consent Statement:** Not applicable.

**Data Availability Statement:** The data that support the findings of this study are available from the author upon reasonable request.

**Conflicts of Interest:** The authors declare no conflict of interest.

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
