# Peer review of "Research on the Estimation of Chinese Fir Stand Volume Based on UAV-LiDAR Technology"

_forests, doi:10.3390/f14061252_

Round 1

Reviewer 1 Report

This study introduced a method to estimate the volume of Chinese Fir based on UAV-LiDAR. The background of this topic has been clearly provided in the introduction. The method and results have also been explained in detail.  The result reaches a high accuracy of 90.5%. Overall, the paper is well-written and under the scope of forests. It can be accepted after the following revisions:

The airborne LiDAR has a limitation in capturing vertical information. For example, the area under the tree canopy may be inaccurate due to the obstruction by the surrounding vegetation. I think it is worth mentioning the limitation in the discussion to make the argument more rigorous.

There are many typos, such as “(1. 2. Study Area” in page 2, “i. 3. Methodology and scheme design” in page 3.

The right images in Figure 1 can be enlarged.

Figure 3 is not clear. What do the three images mean? Please add captions.

Table 1 provides some features, such as area, length and compactness. Some features are missing, such as diversity and connectivity. Please see the following reference for more details: "Development and application of 3D spatial metrics using point clouds for landscape visual quality assessment." Landscape and Urban Planning (2022). This can also be the limitations or discussion section.

Some figures and tables can be combined, such as Figures 5-7.

A  traditional paper structure consisting of an introduction, method, results, discussion and conclusion is recommended.

Minor editing of English language required

Author Response

Thanks for your valuable suggestion, I have uploaded a PDF version of the response.

Reviewer 2 Report

Forests - Review of Manuscript

Subject: Manuscript ID:  Forests – 2418835 “Research on the Estimation of Chinese Fir Stand Volume Based on UAV-LiDAR Technology”

Overall summary:

This manuscript is a study on the use of UAV-Lidar data to automate the process of identifying three forms of ‘recognition units’, defined as clusters of either 1, 2 or 3 Chinese fir trees, which should provide more accurate estimates of stand volume compared to using single tree segmentation algorithms. A classification model was developed to identify the different three ‘recognition unit’ forms based on lidar derived measures of the geometric features of sample units. Separately, for each form of ‘recognition unit’, volume prediction models were developed using lidar derived metrics of combined crown area and average height of trees within the ‘recognition unit’ as predictor variables. The classification and unit volume prediction models were then applied to an independent set of plots to test the accuracy of the models to quantify stand level volume. Present of results need significant improvement.

Specific items:

The purpose of the study is clear, that identifying different forms of Chinese fir should be an improvement over using single tree segmentation for estimating stand volume. However, I have two major concerns: (1) they did not compare their results to using only single tree segmentation; and (2) the presentation of the methods and results is not suitable for publication.

Specifically, the section numbers are incorrect throughout the manuscript, and many figures and tables are not referenced in the text. Figure captions generally do a poor job of defining the content in the figures. Table captions should be ‘Table XX’, not ‘Tab XX’. Results contain duplicate presentation of same data, unnecessary graphs should be removed. 

I was constantly confused between their use of STAND versus tree volume models, e.g., they refer to screening ‘Stand’ volume models based on DBH and HT as predictors (pg. 7), but these are really individual tree models, not stand models. Similarly, they are trying to predict volume within ‘recognition units’ using crown area and height as predictors, so it is not a ‘stand’ volume. Another instance is on page 9, equation 2, where they state dependent variable V is stand level volume, while predictors are based on single tree segmentations.

The sections are in the wrong order for the reader to understand the methods. I had many questions about details of their methods, only to find the information was provided in a later section. The order should be (1) Introduction; (2) Study Area; (3) Data Acquisition and Processing; (4) Remote Sensing Analysis (instead of Methodology and scheme design); (5) Results; (6) Discussion; and (7) Conclusion.

Many references in citation list do not conform to journal instructions
(https://mdpi-res.com/data/mdpi_references_guide_v5.pdf).

I have made numerous other specific comments in the PDF file of the manuscript, which I attached to this submission. 

Many terms they were used are not consistent with common forestry terminology. 

Author Response

(The authors gave the same response as above.)
